# The Development of Smart Dairy Farm System and Its Application in Nutritional Grouping and Mastitis Prediction

**DOI:** 10.3390/ani13050804

**Published:** 2023-02-23

**Authors:** Tingting Hu, Jinmen Zhang, Xinrui Zhang, Yidan Chen, Renlong Zhang, Kaijun Guo

**Affiliations:** 1College of Animal Science and Technology, Beijing University of Agriculture, Beijing 102206, China; 2Department of Computer and Information Engineering, Beijing University of Agriculture, Beijing 102206, China

**Keywords:** smart dairy farm system, nutritional grouping, greenhouse gas reduction, mastitis prediction, dairy cows

## Abstract

**Simple Summary:**

This study combined Internet of Things technology with dairy farm management to set up a smart dairy farm system (SDFS). All kinds of data in the dairy farm will be intelligently captured by various sensors and transmitted to the SDFS in time for corresponding integration analysis. Nutritional grouping was demonstrated to improve production performance and methane and carbon dioxide emission reduction, which is also a hotspot of concern for the public and scientific research. The information from dairy herd improvement (DHI) analysis was used to predict the incidence of mastitis in dairy cows, which would lead to a new way to predict individual mastitis. By fully interpreting the hidden value of dairy farm data, SDFS could help in the better management of dairy farms and promote the application of intelligent systems in dairy farm production.

**Abstract:**

In order to study the smart management of dairy farms, this study combined Internet of Things (IoT) technology and dairy farm daily management to form an intelligent dairy farm sensor network and set up a smart dairy farm system (SDFS), which could provide timely guidance for dairy production. To illustrate the concept and benefits of the SDFS, two application scenarios were sampled: (1) Nutritional grouping (NG): grouping cows according to the nutritional requirements by considering parities, days in lactation, dry matter intake (DMI), metabolic protein (MP), net energy of lactation (NEL), etc. By supplying feed corresponding to nutritional needs, milk production, methane and carbon dioxide emissions were compared with those of the original farm grouping (OG), which was grouped according to lactation stage. (2) Mastitis risk prediction: using the dairy herd improvement (DHI) data of the previous 4 lactation months of the dairy cows, logistic regression analysis was applied to predict dairy cows at risk of mastitis in successive months in order to make suitable measurements in advance. The results showed that compared with OG, NG significantly increased milk production and reduced methane and carbon dioxide emissions of dairy cows (*p* < 0.05). The predictive value of the mastitis risk assessment model was 0.773, with an accuracy of 89.91%, a specificity of 70.2%, and a sensitivity of 76.3%. By applying the intelligent dairy farm sensor network and establishing an SDFS, through intelligent analysis, full use of dairy farm data would be made to achieve higher milk production of dairy cows, lower greenhouse gas emissions, and predict in advance the occurrence of mastitis of dairy cows.

## 1. Introduction

With the rapid development of modern science and technology and the continuous improvement of communication technology, the development and application of the Internet of Things (IoT) have gradually been penetrating every aspect of life [1]. It is estimated that there would be about 2.5 billion bytes of data every day, which is beyond the capacity of human computing. Artificial intelligence (AI), big data, and machine learning (ML) emerged as the times require [2]. Traditional agriculture has gradually entered the era of intelligent agriculture after the era of mechanization and automation [3]. Smart animal husbandry combines farm management with technologies such as IoT, 5G, ML, and AI to improve the production level of animal husbandry and promote profound changes in animal husbandry [4,5]. Smart dairy farming is a specific application of animal husbandry intelligence, and it is also the mainstream trend in the modernization of dairy farms. Nowadays, ever more dairy farms are covered with large areas of 5G networks and sensors [6]. Precise feeding is performed regularly and quantitatively using automated technology [7], and dairy farms are managed automatically and intelligently using technologies such as automatic ventilation and drainage [8] and air quality detection [9]. Dairy farms generate more and more data while the dairy farm scale expands [10]. For example, radio frequency identification technology is used to recognize individual dairy cows and obtain information such as the location and body temperature [11]. Different models (Wood model, Ali Schaeffer model, etc.) are used to predict 305d milk yield of cows and then evaluate the effect of genetic evaluation of dairy cows [12]. The movement of dairy cows can be precisely calculated by a collar monitoring system to reveal whether animals are in heat [13]. Different data streams represent different information, and most have their own collection and analysis systems. Most dairy farms are not able to effectively utilize data streams from different sources to extract their full value. Determining how to effectively integrate and utilize these data streams for dairy farm management has become a challenge [14]. Studies have shown that the management of the entire farm can be improved by integrating these data streams in real time, which in turn enables data-driven decision making on the farm [15,16].

In China, almost every large dairy farm has 2–3 management systems. The levels of dairy farm management systems are uneven and there is no well-recognized system that can efficiently integrate different sources of data flow and propose corresponding improvement measures for dairy farms. In order to solve this problem, we developed a system for data collection, processing, and analysis, namely a smart dairy farm system (SDFS), and illustrated the concept of the SDFS through two application scenarios: (1) nutritional grouping of dairy cows through cluster analysis for precise nutritional management, improving milk production and reducing greenhouse gas (GHG) emissions; (2) risk prediction of dairy cow mastitis through logistic regression to identify the dairy cows with mastitis risk in order to take measurements in advance. Through the overall analysis of dairy farm data, the SDFS could make better use of dairy farm data and improve the management level of dairy farms. However, farm data are a huge treasure, and the data used in this paper are only a part of the huge data of farms, so we need to constantly excavate, analyze, and interpret the data behind the farms and constantly promote the application of intelligent systems in farm production management.

## 2. Materials and Methods

### 2.1. SDFS Establishment and Data Collection

#### 2.1.1. Setup of SDFS

A dairy farm in the Beijing area with a herd of 2500 cows, including 1256 lactating cows, was selected. The dairy farm is designed with an intelligent sensor network structure, and dairy farm managers can achieve remote and precise monitoring and control via computer and mobile phone. The dairy farm was equipped with a large number of diversified intelligent devices for cow monitoring and management, such as radio frequency identification (RFID) electronic ear tags [17]. A weighing system is combined with RFID technology to obtain the weight information of individual cows, which is sent to a management computer, providing management staff with basic information for the management of cattle farms. The farm also has a total mixed ration (TMR) precision feeding system [18] and unmanned aerial vehicle (UAV) photography system [19]. The installation of sensors for monitoring temperature, humidity, wind speed, and GHGs (such as methane (CH_4_) and carbon dioxide (CO_2_)) [20] enables the monitoring of environmental parameters in the dairy farm (Figure 1). The dairy farm data are transmitted to the computer terminal of the SDFS (as a database) through sensors, and the relevant data can be extracted from the database for advanced analysis and visual presentation. The system enables the timely collection and analysis of data collected by the dairy farm sensor network. The SDFS includes automatic monthly reports, annual reports, growth performance, DHI data, reproductive performance, individual cow value display and analysis, nitrogen and phosphorus emissions, and GHG emissions. Two application examples (nutrient grouping and mastitis prediction) ae used below to illustrate the application of the SDFS.

#### 2.1.2. Data Collection in Dairy Farm

The data collected mainly included the following: (1) individual information: cattle pedigree information, body weight, appearance score, body condition score, etc.; (2) farm information: farm location, stalls, cattle barns, environment, etc.; (3) cattle management: recording routine events in herds, such as heat, insemination, calving, and disease prevention and control events; (4) feed: including diet composition, average delivery and feed residues per day, and dry matter intake (DMI); (5) dairy herd improvement (DHI) parameters: milk yield, milk fat percentage, milk protein percentage, fat/protein ratio, milk fat content, milk protein content, and somatic cell count (SCC) were recorded on monthly test days [21].

### 2.2. Application Scene of Smart Dairy Farm System

#### 2.2.1. Nutrition Grouping

Based on the individual cow information, DMI, and DHI data collected by the SDFS, Nutrient Dynamic System Professional Software (NDS, developed by Rum&n srl), which adopted the Cornell Net Carbohydrate and Protein System (CNCPS6.55) [22], was used to calculate metabolic protein (MP) and metabolic energy (ME) as two of references for nutritional grouping. Two hundred seventy lactating cows were divided into 9 pens by cluster analysis as the nutritional group (NG). The control groups were assigned to 9 pens with 30 cows in each pen according to the original grouping (OG; divided according to milk production of cows) method in the farm, in which cows were grouped according to lactation stages. The corresponding diets calculated by NDS were provided for each pen (Table 1 and Table 2). Since the monitoring of methane and carbon dioxide is greatly influenced by the environment, space, and air flow, the stability of the data detected by this sensor needs to be further improved. Therefore, the methane and carbon dioxide data in this study mainly used the predicted value of CNCPS system, and the stability was guaranteed to a certain extent [23]. N intake was calculated and CH_4_ and CO_2_ emissions were also estimated by NDS [24].

#### 2.2.2. Mastitis Prediction

The mastitis prediction was carried out using the original DHI records of the dairy farm from 1 January 2019 to 31 December 2021. In general, SCC greater than 200,000/mL was used as the criterion for subclinical mastitis in dairy cows [25]. From the original DHI records, records without SCC were deleted. Due to the skewed distribution of the SCC values and the heterogeneous variance, SCC was first transformed into somatic cell score (SCS), which was close to a normal distribution, before the subsequent analysis [26]. The conversion formula was as follows:SCS = log2(SCC/100000) + 3.

To improve the accuracy of model predictions, the values of SCS ranging from 0 to 10 were used [27]. According to the SCC values, cows were divided into a healthy group and a mastitis risk group (Table 3). According to the SCC in DHI data, when SCC ≤ 200,000/mL, it indicates a healthy condition; when 200,000/mL < SCC ≤ 500,000/mL, it indicates subclinical mastitis, when SCC > 500,000/mL, it indicates clinical mastitis [28]. In this paper, both subclinical and clinical mastitis were named as the mastitis risk group.

Altogether, 2555 DHI records were used for regression analysis. The following independent variables were chosen: parity, days in milk (DIM), and milk indicators of the previous four lactation months (milk yield, milk protein percentage, milk fat percentage, lactose percentage, fat/protein ratio). A training set (70%) and a validation set (30%) were created from the collected data. The training set was used to filter the independent variables (equation 1) by bidirectional elimination stepwise regression. The parameters with statistically significant effects on the prediction of mastitis in dairy cows were obtained (Table 4), and corresponding coefficients were substituted into the logistic regression equation for further analysis using the validation set:(1)$$\mathrm{logit}\left(P\right)=\log\left(\frac{P}{1-P}\right)=\beta_{0}+\beta_{1}X_{1}+\beta_{2}X_{2}+\beta_{3}X_{3}+\beta_{4}X_{4}+...\beta_{26}X_{26}$$
where *P* represents the probability of positive results; 1 − *P* represents the probability of non-positive results. $\frac{P}{1-P}$ is the strength of the disease as a statistical indicator, called the odds ratio (OR), used to estimate the effect of the independent variable on disease. X_1_ is parity, X_2_–X_5_ represent the amount of milk produced during the first 1–4 lactation months, X_6_–X_9_ represent milk fat percentage in the first 1–4 lactation months, X_10_–X_13_ represent the protein rate of the first 1–4 lactation months, X_14_–X_17_ represent lactose rate in the first 1–4 lactation months, X_18_–X_21_ represent the fat-to-egg ratio of the first 1–4 lactation months, X_22_–X_26_ represent the natural months of the first 1–5 lactation months; β_0_ is a constant, and β_1_–β_26_ are the regression coefficients of each variable (X_1_–X_26_).

The receiver operating characteristic (ROC) curve was plotted to reflect the prediction accuracy of predictive variables in the dairy cow mastitis risk assessment model. The *X*-axis is specificity (percentage of healthy cows that tested negative) and the *Y*-axis is sensitivity (the percentage of risk cows that correctly tested positive), The area under the curve (AUC) was calculated. An AUC of the prediction model between 0.50 and 0.70 indicates that the effect is average, an AUC between 0.70 and 0.90 is considered to indicate a good model, and an AUC higher than 0.90 is considered to indicate an excellent model [29,30].

### 2.3. Statistical Analysis

First of all, the data were preliminarily sorted using Excel and SPSS 21.0.

Cluster analysis was performed using the hclust function in the stats package of Rx64 (version 4.0.5) for nutritional parameters (milk production, parity, DIM, MP, ME, etc.).

For mastitis prediction, the DHI data were analyzed using ANOVA in SPSS 21.0. Logistic regression analysis was performed using the general linear model (GLM) function, and the ROC curve was drawn using the PROC package of Rx64 software.

## 3. Results

### 3.1. Application of SDFS

#### 3.1.1. Nutrient Grouping

As shown in Table 5, compared with OG, the milk production of NG increased significantly (*p* < 0.05), except that the milk production in the mid-lactation group of the second parity was not significantly different. In general, the use of NG can significantly improve the milk yield of dairy cows at the same stage.

As shown in Table 6, the N intake of dairy cows in NG was higher than that in OG; N production and N efficiency in NG were highly significantly increased than that in OG (*p* < 0.01). Compared to OG, overall N efficiency in NG increased by 1.98%. Generally speaking, the N intake of cows increased after NG treatment.

The GHG emissions are shown in Table 7. CH_4_ and CO_2_ emissions of dairy cows in NG were lower than those in OG. In general, the use of NG resulted in a decrease in dairy cow methane and carbon dioxide emissions.

#### 3.1.2. Mastitis Prediction

The risk factors related to mastitis in the experimental dairy farm are shown in Table 8. OR values indicated the fitting degree of the model was good (Hosmer–Lemeshow (*p* > 0.05)). The results showed that milk yield in the second lactation month (*p* < 0.05), fat percentage in the first and third lactation months (*p* < 0.05), and natural month in the fifth lactation month (*p* < 0.05) had significant effects on mastitis risk in dairy cows. The predictive value of the mastitis risk assessment model was 0.773. The accuracy was 89.9%, the specificity was 70.2%, and the sensitivity was 76.3% (Figure 2).

## 4. Discussion

Recently, the utilization of 5G and IoT technology has become a major trend in the development of animal husbandry [31]. This study formed a large sensing network by applying the concept of precision animal husbandry and installing various types of equipment on dairy farms. Data are collected through various sensors, and IoT facilitates the transmission of data from the network to the SDFS for data analysis, forming the basis for the transformation of intelligent dairy farms into smart management.

### 4.1. Nutrition Grouping

Currently, most dairy farms usually only consider the lactation stage when grouping cows to determine feed ration, resulting in less accurate feed nutrition on dairy farms [32]. Studies have shown that grouping dairy cows according to their actual nutritional needs can increase the utilization rate of N in the diet [33], and milk production, so as to reduce GHG emissions [34].

In this study, for dairy cows in NG, milk production significantly increased (*p* < 0.05). The dairy cows with different needs were fed different TMR formulations, which greatly improved the N efficiency and increased milk production. This is in agreement with the results of Cabrera et al. [35]. Nutritional grouping of dairy cows could improve the nutritional accuracy of diet, prevent nutrient loss, reduce dietary costs, and increase milk production.

Nutritional grouping had better theoretical nutritional accuracy, thereby reducing nutritional loss due to dietary and environmental influences [36]. GHG emissions (CH_4_, N_2_O, CO_2_) are currently a research hotspot all over the world [37,38]. In this study, dairy cows in NG had CH_4_ and CO_2_ emissions lower than those in OG, and the N efficiency increased by 1.98% on average. Kalalantari et al. [39]. reported an increase in N efficiency by 2.7% when cows were divided into multiple nutritional groups and fed with an average MP lower than requirements. The methane emission also decreased. The reason for this discrepancy could be that the authors fully considered the cow’s body weight (BW), BCS (the range of 2.0–4.5 to ensure the accuracy of the model), and NEL in their study, while in this study these factors were not in consideration. Therefore, the nutritional grouping of lactating dairy cows can fully consider the physical condition and nutritional needs of dairy cows and provide different diets for different groups to better meet the nutritional needs of dairy cows and lower GHG emissions [40].

### 4.2. Smart Prediction of Mastitis

As dairy cows produce ever more milk, cow mastitis has been becoming one of the most important diseases that restrict the development of the global dairy industry [41]. It is estimated that economic losses due to mastitis in dairy cows account for 38% of the total direct cost of common production diseases on dairy farms [42]. Risk assessment and timely prevention of mastitis in cows are essential to ensure the health of cows and the consistent improvement of raw milk quality [43].

Recent studies on mastitis prediction have focused on factors such as year, month, and farm [44,45]. Individual information about cows has not been taken into account, making the prediction less practical. This study screened the factors that could predict the mastitis risk in the fifth lactation month based on the previous four lactation months using DHI data of individual cows. The milk yield in the second lactation month and fat percentage in the first and third lactation months were influential risk factors for mastitis and could accurately predict the risk of mastitis in the fifth lactation month. In this way, the incidence of mastitis in the dairy farm could be predicted two months in advance. This is expected to have a certain guiding significance for dairy farms.

The ROC curve is a comprehensive representation of a model’s accuracy. Sensitivity and specificity are indispensable indicators that reflect authenticity [46]. The area under the curve (AUC) is the diagnostic value of the model. The larger the area is, the better the diagnostic performance of the model is [47]. The predictive specificity of this study was similar to Cavero’s predictive specificity (74.9%), which was calculated for mastitis prediction using data from 478 cows from neural networks and automated milking systems [18]. Sun et al. [48] reported a result of 87% true positives using artificial neural networks, which was similar to this study. The predictive value of this study was lower than the predictive value (AUC) of 0.93 reported by Jadhav et al. [49] based on a 214-cow dataset. This may be due to the authors using more variables such as mammary and bedding hygiene and milking methods in their study. In practice, bedding hygiene status, teat shape, and udder hygiene also affect the occurrence of mastitis in dairy cows, interfering with the accuracy of individual dairy cow mastitis risk assessment [50].

However, the prediction process could be always developing. With continuous accumulation and aggregation of more data and real-time data streams, the accuracy of model prediction could be improved over time. According to logistic regression and ROC curve description, we can find the risk indicators of dairy mastitis. Significantly, this equation does not apply to all cases; our team is still studying further, hoping to find a general equation that can be updated automatically by continuously merging the past data to detect the incidence of cow disease in time.

## 5. Conclusions

This study combined IoT technology with dairy farm management to set up an SDFS. All kinds of data in the dairy farm will be intelligently captured by various sensors and transmitted to the SDFS in time for corresponding integration analysis. The applications of the SDFS were demonstrated in two aspects. NG according to the nutritional needs of dairy cows could improve nutritional accuracy, thus leading to increased milk production and N efficiency and reduced CH_4_ and CO_2_ emissions, so as to mitigate the environmental impacts. A mastitis prediction model was established using DHI data to identify potential cows at risk of mastitis in advance, thus reducing economic losses. By fully interpreting the hidden value of dairy farm data, the SDFS could help in the better management of dairy farms and promote the application of intelligent systems in dairy farm production. At present, our data mining of dairy farms is not comprehensive enough; we still need to continue to work hard.

## Figures and Tables

**Figure 1 animals-13-00804-f001:**
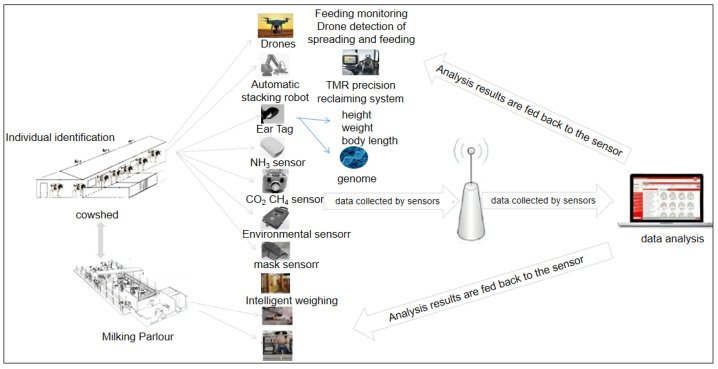
Structure of smart dairy farm sensing network.

**Figure 2 animals-13-00804-f002:**
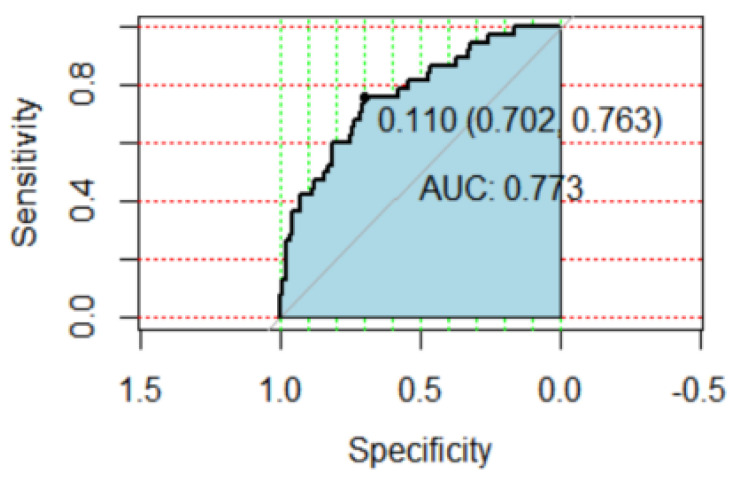
ROC curve of mastitis prediction.

**Table 1 animals-13-00804-t001:** Diet composition and routine nutrient composition—original farm grouping.

Ingredient	TMR Diet Formulation (kg) Pens (OG)				
	1	2 and 3	4 and 7	5 and 8	6 and 9
Corn silage	23.00	23.00	23.31	23.32	23.21
Alfalfa hay	2.34	2.20	2.31	2.40	2.31
Oat hay	2.38	2.18	2.39	2.50	2.39
Corn grain fine	3.10	2.70	3.35	2.92	3.21
Soybean meal	2.93	2.66	3.32	3.00	3.12
Soft wheat bran	1.294	1.263	1.45	1.38	1.32
Soybean steam flaked	1.280	1.291	1.44	1.38	1.32
Beet pulp pellet	1.287	1.312	1.46	1.37	1.36
Calcium salt of fatty acids	0.10	0.00	0.40	0.10	0.15
Sugarcane molasses	0.51	0.35	0.60	0.30	0.56
Cottonseed meal	0.50	0.40	0.62	0.30	0.60
Premix	0.40	0.55	0.45	0.55	0.40
Nutritional composition					
ME (Mcal/day)	56.29	52.87	61.51	56.21	58.28
MP (g/day)	2518.80	2344.80	2732.50	2484.10	2604.02
CP (%)	16.76	16.46	17.10	16.63	17.02
Crude fat (%)	4.00	3.69	4.97	4.04	4.14
NFC (%)	37.85	37.32	37.34	37.02	37.80
NDF (%)	33.98	34.49	32.97	34.30	33.65
peNDF (%)	24.77	25.23	24.03	25.01	24.23
Starch (%)	24.26	24.15	24.24	23.92	24.14
Predicted DMI (kg/cow per day)	21.66	20.60	23.30	21.81	22.60

1–3 stand for diets for early, middle, and late stages of first lactation, respectively; 4–6 stand for early, middle, and late stages of second lactation, respectively; 7–9 stand for early, middle, and late stages of third lactation and over, respectively.

**Table 2 animals-13-00804-t002:** Diet composition and routine nutrient composition—nutrient grouping.

Ingredient	TMR Diet Formulation (kg) Pen (NG)								
	1	2	3	4	5	6	7	8	9
Corn silage	22.00	22.15	22.00	22.20	22.20	22.30	22.60	22.43	22.41
Alfalfa hay	2.20	2.20	2.00	2.21	2.21	2.31	2.40	2.28	2.28
Oat hay	2.00	2.00	2.00	2.34	2.34	2.32	2.52	2.22	2.28
Corn grain fine	3.32	3.14	2.84	3.54	3.43	3.22	3.66	3.57	3.36
Soybean meal	3.34	3.20	2.87	3.54	3.44	3.24	3.69	3.58	3.34
Soft wheat bran	1.17	1.20	1.27	1.57	1.37	1.16	1.44	1.38	1.36
Soybean steam flaked	1.13	1.11	1.27	1.43	1.23	1.13	1.54	1.37	1.27
Beet pulp pellet	1.30	1.30	1.21	1.40	1.20	1.11	1.46	1.36	1.26
Calcium salt of fatty acids	0.40	0.25	0.10	0.44	0.33	0.25	0.50	0.43	0.25
Sugarcane molasses	0.62	0.60	0.40	0.62	0.60	0.60	0.60	0.56	0.56
Cottonseed meal	0.58	0.50	0.40	0.60	0.54	0.58	0.60	0.60	0.60
Premix	0.41	0.45	0.40	0.43	0.43	0.35	0.40	0.38	0.38
Nutritional composition of diet									
ME (Mcal/day)	57.85	55.90	52.82	61.55	58.99	57.01	63.75	61.76	58.62
MP (g/day)	2572.00	2491.80	2336.40	2764.50	2630.60	2526.70	2863.90	2766.10	2622.80
CP (%)	17.25	17.07	16.98	17.46	17.28	17.18	17.56	17.57	17.37
Crude fat (%)	4.91	4.39	4.09	5.11	4.71	4.43	5.28	5.06	4.46
NFC (%)	38.01	38.04	37.83	37.52	37.90	38.05	37.33	37.74	37.90
NDF (%)	32.19	32.77	33.63	32.35	32.52	33.00	32.36	32.26	32.91
peNDF (%)	22.93	23.52	24.28	22.44	23.29	24.26	22.67	22.65	23.51
Starch (%)	24.47	24.37	24.59	23.96	24.28	24.23	23.70	24.25	24.29
Predicted DMI (kg/cow per day)	22.14	21.64	20.73	23.64	22.88	22.30	24.14	23.47	22.92

1–3 stand for diets for early, middle, and late stages of first lactation, respectively; 4–6 stand for early, middle, and late stages of second lactation, respectively; 7–9 stand for early, middle, and late stages of third lactation and over, respectively.

**Table 3 animals-13-00804-t003:** Disease grouping of Holstein cows.

Group	SCC ^1^, 10^4^/mL	SCS ^2^
Health Group	SCC ≤ 20	SCS ≤ 4
Risk Group	SCC > 20	SCS > 4

^1^ SCC: somatic cell count. ^2^ SCS: somatic cell score.

**Table 4 animals-13-00804-t004:** Risk factors of mastitis in Chinese Holstein cattle in the Beijing area.

Variable	Abbreviation	Name
State of illness	Health	0
	Risk	1
Milk yield 2	cnl2	X_3_
Fat percentage 1	rzl1	X_6_
Fat percentage 2	rzl2	X_7_
Fat percentage 3	rzl3	X_8_
Protein percentage 1	dbl1	X_10_
Lactose percentage 4	rtl4	X_17_
Fat/protein ratio1	zdb1	X_18_
Fat/protein ratio 3	zdb3	X_20_
Month 5	yuefen5	X_26_

**Table 5 animals-13-00804-t005:** Effects of farm group and nutrient group strategies on milk production of dairy cows.

Parity	Pen	Stage	Milk yield		*p* Value
			OG ^1^	NG ^2^	
1	1	Early	37.45 + 3.45 ^a^	39.28 + 2.90 ^b^	0.031
	2	Mid	35.78 + 4.22 ^a^	37.87 + 3.19 ^b^	0.035
	3	Late	33.10 ±2.90 ^a^	34.84 ±2.33 ^b^	0.013
2	4	Early	40.17 ±4.07 ^a^	42.29 ± 2.98 ^b^	0.025
	5	Mid	38.34 ± 4.22	39.74 ± 3.47	0.165
	6	Late	36.48 ± 3.70 ^a^	38.20 ± 2.75 ^b^	0.046
≥3	7	Early	41.22 ± 4.36 ^a^	43.58 ± 3.83 ^b^	0.030
	8	Mid	39.20 ± 3.81 ^a^	41.64 ± 4.55 ^b^	0.029
	9	Late	37.55 ± 3.38 ^a^	39.31 ± 3.14 ^b^	0.041

Different lowercase letters in the same row differ significantly (*p* < 0.05). ^1^ OG: original grouping. ^2^ NG: nutritional grouping.

**Table 6 animals-13-00804-t006:** Effects of different grouping strategies on dietary N utilization.

Parity	Pen	Stage	N Intake (g)		NG- OG	N Production (g)		*p* Value	N Efficiency (%)		*p* Value	Changes in N Efficiency ^3^ (%)
			OG ^1^	NG ^2^		OG	NG		OG	NG		
1	1	Early	598.88	617.10	18.22	201.94 ± 0.90 ^B^	216.19 ± 0.76 ^A^	<0.001	33.72 ± 0.15 ^B^	35.03 ± 0.12 ^A^	<0.001	3.88
	2	Mid	579.68	597.89	18.21	197.86 ± 1.06 ^B^	210.80 ± 1.06 ^A^	<0.001	34.13 ± 0.18 ^B^	35.26 ± 0.18 ^A^	<0.001	3.31
	3	Late	546.03	565.11	20.50	184.87 ± 0.90 ^B^	194.38 ± 1.16 ^A^	<0.001	33.86 ± 0.17 ^B^	34.40 ± 0.20 ^A^	<0.001	1.59
2	4	Early	654.1	666.13	12.03	224.40 ± 0.70 ^B^	233.28 ± 0.83 ^A^	<0.001	34.31 ± 0.11 ^B^	35.02 ± 0.12 ^A^	<0.001	2.07
	5	Mid	624.9	634.84	9.94	215.92 ± 1.07 ^B^	219.03 ± 1.07 ^A^	<0.001	34.55 ± 0.17	34.50 ± 0.17	0.248	−0.14
	6	Late	598.03	610.95	12.92	203.08 ± 1.01 ^B^	212.16 ± 0.95 ^A^	<0.001	33.96 ± 0.17 ^B^	34.73 ± 0.15 ^A^	<0.001	2.27
≥3	7	Early	677.66	691.39	13.73	231.61 ± 1.09 ^B^	237.45 ± 1.39 ^A^	<0.001	34.18 ± 0.16 ^B^	34.37 ± 0.15 ^A^	<0.001	0.56
	8	Mid	645.23	663.15	17.92	218.02 ± 0.96 ^B^	228.40 ± 1.99 ^A^	<0.001	33.79 ± 0.15 ^B^	34.49 ± 0.15 ^A^	<0.001	2.07
	9	Late	623.76	635.36	11.6	210.00 ± 0.96 ^B^	217.73 ± 1.53 ^A^	<0.001	33.67 ± 0.15 ^B^	34.40 ± 0.14 ^A^	<0.001	2.17

Different uppercase letters in the same row differ highly significantly *(p* < 0.01). ^1^ OG: original grouping. ^2^ NG: nutritional grouping. ^3^ Changes in N efficiency = (NG–OG)/OG.

**Table 7 animals-13-00804-t007:** Methane and carbon dioxide emissions from cattle for group and nutrient group strategies.

Parity	Pen	Stage	CH_4_, g/d/Head		Difference (%)	CO_2_, kg/d/Head		Difference (%)
			OG ^1^	NG ^2^		OG	NG	
1	1	Early	451.46	450.78	−0.15	13.16	13.14	−0.15
	2	Mid	445.10	443.55	−0.35	12.97	12.95	−0.15
	3	Late	432.10	428.40	−0.86	12.51	12.46	−0.40
2	4	Early	474.76	472.24	−0.53	13.90	13.82	−0.58
	5	Mid	461.90	459.06	−0.61	13.50	13.46	−0.30
	6	Late	453.69	452.21	−0.33	13.23	13.10	−0.98
≥3	7	Early	486.24	485.58	−0.14	14.26	14.18	−0.56
	8	Mid	475.23	469.33	−1.24	13.86	13.72	−1.01
	9	Late	463.48	458.97	−0.97	13.56	13.39	−1.25

^1^ OG: original grouping. ^2^ NG: nutritional grouping.

**Table 8 animals-13-00804-t008:** Risk factors of mastitis.

Variable	Abbreviation	Β ^1^	OR ^2^	95%CI ^3^		*p* Value
Milk yield 2	cnl2	0.14	1.15	1.01	1.31	0.031
Fat percentage 1	rzl1	1.90	6.72	0.93	48.79	0.036
Fat percentage 2	rzl2	0.95	2.60	0.77	8.71	0.127
Fat percentage 3	rzl3	1.20	3.32	1.41	7.81	0.003
Protein percentage 1	dbl1	−2.73	0.07	0.00	1.46	0.054
Lactose percentage 4	rtl4	−4.02	0.02	0.00	0.15	0.000
Fat/protein ratio 1	zdb1	−4.53	0.01	0.00	4.16	0.102
Fat/protein ratio 3	zdb3	−2.55	0.08	0.01	1.19	0.043
Month 5	yuefen5	0.14	1.15	1.02	1.30	0.021

^1^ β = regression coefficient. ^2^ OR = odds ratio. ^3^ CI = confidence interval.

## Data Availability

The data presented in this study are available on request from the corresponding author. The data are not publicly available.

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
