# Peer review of "The Development of Smart Dairy Farm System and Its Application in Nutritional Grouping and Mastitis Prediction"

_animals, 2023, doi:10.3390/ani13050804_

Round 1

Reviewer 1 Report

The title of the paper would be better adapted to the content, i.e. to the control of a demand-oriented and environmentally friendly feeding of dairy cows and the prediction of the risk of mastitis. In Material and Methods, the entire sensor network available on the farm is described. However, the reader must then realise that only a small part is used. The fact must be explained beforehand. It is important, however, that e.g. the formation of cow groups according to nutrient requirements is described in more detail. This also includes showing the variance in nutrient requirements within the groups. It is recommended that the conclusions drawn from the study results should be more comprehensive.

Author Response

Reviewer  Report

The title of the paper would be better adapted to the content, i.e. to the control of a demand-oriented and environmentally friendly feeding of dairy cows and the prediction of the risk of mastitis. In Material and Methods, the entire sensor network available on the farm is described. However, the reader must then realise that only a small part is used. The fact must be explained beforehand. It is important, however, that e.g. the formation of cow groups according to nutrient requirements is described in more detail. This also includes showing the variance in nutrient requirements within the groups. It is recommended that the conclusions drawn from the study results should be more comprehensive.

Response:

Thanks for your suggestion, The title has been changed: The development of Smart Dairy Farm System and its application in nutritional grouping and mastitis prediction.

The facts that “only a small part is used” was explained in the article (pls see L.81-83). The sensor mentioned in MM is used as part of data resources of SDFS, other data resourses could be laboratory analysis, staff registeration, etc. SDFS would collect all kind of data and generate more sophisticated solution for improving farm management.

The results and discussion, have been refined. Such as, Nutritional grouping of dairy cows can better meet the daily application requirements of dairy cows, which can improve the nutritional accuracy of diet, prevent nutrient loss, reduce dietary costs and increase milk production (pls see L.243).

Reviewer 2 Report

Large number of data produced in dairy farm as the rapidly development and application of IoT. However, these data are underused to generate due data benefits as it should be. This manuscript setup a smart dairy farm system to integrating these data streams, and gave examples on the application scenarios of nutritional grouping and mastitis risk prediction to show the benefits from fully data utilization. This gives evidence that farmer can benefit from better management and enhances our confidence in using IoT and precision farming technology to promote the application in dairy production. There are in-text errors need to be corrected, for example the decimal symbol in table 8.

Line 147-153: keep the variables consistent with those in table 4. For example, X1 is parity in line 147, while x1 is milk yield 2 in table 4.

Table 4: what’s the difference between fat percentage 1 to fat percentage 3? Please explain the meaning of the variables in table 4, e.g. what does “Month 5” represent?

Line 170: delete the comma. Check the full text carefully to make sure the commas are correctly used. for example, a comma is needed after “recently” in line 224.

Table 5: the nutrient group had higher ME, CP in diet and higher feed intake as show in table 1 and table 2, how about the feed cost and feed to milk ratio?

Line 173: explain what UAV represent for when it first appears.

Table 6-table 7: I am very curious on how does the NDS calculate N production, N efficiency and gas emissions, is there any reference about this or could you explain a little bit more?

Line 2224-240: What’s the perspectives and deficiencies need to be improved in the future application of SDFS network? Any consideration about the input-output ratio?

Author Response

Response: First of all, the authors would like to express our thanks to the Academic Editor, and many thanks also to the reviewers for the valuable comments. Give us the chance to improve our manuscript. We really appreciate it.

Please see the following replies to the reviewer’s comments:

Comments and Suggestions for Authors

Large number of data produced in dairy farm as the rapidly development and application of IoT. However, these data are underused to generate due data benefits as it should be. This manuscript setup a smart dairy farm system to integrating these data streams, and gave examples on the application scenarios of nutritional grouping and mastitis risk prediction to show the benefits from fully data utilization. This gives evidence that farmer can benefit from better management and enhances our confidence in using IoT and precision farming technology to promote the application in dairy production.

1.There are in-text errors need to be corrected, for example the decimal symbol in table 8.

Response:Decimal symbol has been revised (pls see L.218).

2.Line 147-153: keep the variables consistent with those in table 4. For example, X1 is parity in line 147, while x1 is milk yield 2 in table 4.

Response: The variables in Table 4 have been corrected in correspondence to those of equation 1 (pls see L.187).

  1. Table 4: what’s the difference between fat percentage 1 to fat percentage 3? Please explain the meaning of the variables in table 4, e.g. what does “Month 5” represent?

Response:The meaning of each variable is explained under equation 1. For example: Milk fat percentage 1 represents milk fat percentage in the first lactation month, Milk fat percentage 3 represents milk fat percentage in the third lactation month, The difference between Milk fat percentage 1 and 3 is the lactation month (pls see L.172-177).

Month 5 represents the natural months of the fifth lactation months, i.e. the month in which mastitis risk is predicted (pls see L.172-177).

4.Line 170: delete the comma. Check the full text carefully to make sure the commas are correctly used. for example, a comma is needed after “recently” in line 224.

Response:Revised and checked full text, Thank you for your suggestion (pls see L.236).

5.Table 5: the nutrient group had higher ME, CP in diet and higher feed intake as show in table 1 and table 2, how about the feed cost and feed to milk ratio?

Response:The manuscript (MS) intended to show that NG was more suitable to dairy cow nutrient requirement, and had better N efficiency. The feed cost and feed to ratio were not in consideration. Anyway, the feed cost would be higher for NG, it will be compensated by higher milk production. Feed to milk ration would be lower with better N efficiency. The author did not put the results in the MS with the intention not putting too much weight on the example.

6.Line 173: explain what UAV represent for when it first appears.

Response:Full spelling has been added to the text. (pls see L.96).

7.Table 6-table 7: I am very curious on how does the NDS calculate N production, N efficiency and gas emissions, is there any reference about this or could you explain a little bit more?

Response:NDS is an commercial software based on the cornell nutrient carbohydrate and protein system (CNCPS) model (pls see L.124). CNCPS is a wide-recognized system designed to predict dairy and beef cattle demand, feed utilization, animal performance, and nutrient excretion based on most advanced research work. It uses a large number of formulas to make dynamic predictions by taking multiple factors into account.

Citations have been added to the text (pls see L.122-135).

8.Line 224-240: What’s the perspectives and deficiencies need to be improved in the future application of SDFS network? Any consideration about the input-output ratio?

Response: Thank you for your comments. The the perspectives and deficiencies of this study have been added to the article (pls see L.296 and 313).

The SDFS would have the input-output ratio in consideration. However, here we just put two examples to show the potential use of SDFS. The input-output ratio which would include many aspects were not discussed in this MS. N efficiency coule be an indicatore of input-output ratio

Reviewer 3 Report

Review animals-2131434

               The title does not correspond with the work done. The title investigates the effect of nutritional grouping and mastitis prediction based on the data from sensors of SDFS. The system is not described in detail (except for some sensors published earlier, but even not cited in the manuscript), as well as the flow of the data through the system. The results achieved through the are also not validated by another system. That is even not discussed. Results are not described enough and the discussion is not correct. Conclusions are not supported enough by the research description. The manuscript needs rewriting.

Specific comments:

L14 – DHI –only shortcut, not clear

L15 – two points

L86 – RFID – not describe, only shortcuts, no citation

L82-94 –this part is not enough for the description of the system. It can stay so if every sensor would have the citation to another paper when it is described and validated

L96-104 – not informative enough, scientific paper need detailed information (or citation to other sources)

L 108-1099 - not informative enough - NDS? CNCP? Give citations at least.

L 118 – not informative enough - DHI?

L 138 – not clear information

L 162-167 should be described with the procedures used.

L 170-184 – not detailed enough

L 183 –two application examples or the structure used? (figure 1)

L 197-206 – results part should not consist only of the tables. The main findings should be described.

L 222- 310 – most parts of this discussion look like an introduction to the subject (l222-235). That is not the place for such overall information. Every part of the discussion starts with an introduction.

Author Response

Response: First of all, the authors would like to express our thanks to the Academic Editor, and many thanks also to the reviewers for the valuable comments. Give us the chance to improve our manuscript. We really appreciate it.

Please see the following replies to the reviewer’s comments:

Comments and Suggestions for Authors

The title does not correspond with the work done. The title investigates the effect of nutritional grouping and mastitis prediction based on the data from sensors of SDFS. The system is not described in detail (except for some sensors published earlier, but even not cited in the manuscript), as well as the flow of the data through the system. The results achieved through the are also not validated by another system. That is even not discussed. Results are not described enough and the discussion is not correct. Conclusions are not supported enough by the research description. The manuscript needs rewriting.

Response: The title has been revised as: The development of Smart Dairy Farm System and its application in nutritional grouping and mastitis prediction.

The description of the system and sensor is also supplemented and cited (pls see L.80-83 and L.86-97).

About the data flow: The farm data from sensors and  laboratories is transmitted to the computer terminal-SDFS (as a database) through sensors, and the relevant data can be extracted from the database for advance analysis and visual presentation (pls see L.99-102 and Figure 1). The validate with another system was not carried out because every system has its own application circumstance and it is not possible to have same situation for comparison. In this manuscript (MS), we used different groups for comparison and validation and model set and validation set in the case of mastitis prediction.

In the results section, we add a summary for the table results (pls see L.199); In the discussion part, we first describe the current situation, then write the similarities and differences between our results and others' results, and discuss the reasons for the differences (pls see L.243).

Specific comments:

1.L14 – DHI –only shortcut, not clear

Response: Full spelling of DHI has been added in the article (pls see L.15).

2.L15 – two points

Response: The second point has been deleted (pls see L.18).

3.L86 – RFID – not describe, only shortcuts, no citation

Response: Full spelling RFID (Radio Frequency Identification) has been added in the article (pls see L.92).

4.L82-94 –this part is not enough for the description of the system. It can stay so if every sensor would have the citation to another paper when it is described and validated

Response: Thanks for your reminding, we have supplemented the literature for each intelligent device as required (pls see L.86-107).

The original intention of our system is to transmit data to the computer system through sensors, and then the system will analyze the data and generate a summary report to better serve the farm. Therefore, the focus of this paper is on system analysis to improve the management and production of farm rather than sensors.

5.L96-104 – not informative enough, scientific paper need detailed information (or citation to other sources)

Response: The data desripted here is common in dairy farm recording and data items were just mentioned. We added citations for clarification (pls see L.111-119).

6.L 108-109 - not informative enough - NDS? CNCP? Give citations at least.

Response: Nutrient Dynamic System Professional Software (NDS, developed by Rum&n srl), which is an analysis software. We input the corresponding dairy cow diet composition into the software, which automatically analyzes the methane carbon dioxide emissions (predicted value) of dairy cows. Citations have been added to the text (pls see L.123-134).

7.L 118 – not informative enough - DHI?

Response: Dairy Herd Improvement (DHI) parameters, including Milk yield, milk fat percentage, milk protein percentage, fat/protein ratio, milk fat content, milk protein content, somatic cell count (SCC) were recorded on monthly test day (pls see L102).

8.L 138 – not clear information

Response: “Description of Equation 1”, the meaning of the paragraph is: we first screen the parameters that have statistical significance on the prediction of cow mastitis through equation 1, and then carry out logistic analysis and ROC curve drawing for these parameters (pls see L158-178).

9.L 162-167  should be described with the procedures used.

Response: Table 4 is at L.162, Table 4 lists the parameters involved in logistic analysis, Statistical analysis: briefly describe the analysis methods used in this paper, We think it is necessary to include these in order to make readers more clear (pls see L.166).

L.163-167 is Statistical analysis; We have refined the description of statistical analysis, for example, we have pointed out what packages we use in R software, etc (pls see L.188-194)

10.L 170-184 – not detailed enough

Response: This paragraph is a simple description of our ranch framework. We have moved it to MM as required and integrated it with the original farm information (pls see L.86-106).

11.L 183 –two application examples or the structure used? (figure 1)

Response: Figure 1: Pasture framework diagram (pls see L.106).

Two application examples (Nutrient grouping and Mastitis prediction) are to show what we are doing with the data collected by the system (pls see L.105).

12.L 197-206 – results part should not consist only of the tables. The main findings should be described.

Response: Thank you very much for your suggestion, we have added a summary description of the main findings of this result based on the original (pls see L.198-209).

13.L 222- 310 – most parts of this discussion look like an introduction to the subject (l222-235). That is not the place for such overall information. Every part of the discussion starts with an introduction.

Response: Thank you for your reminding. We refined the discussion. In the discussion part, we first describe the current situation, then write the similarities and differences between our results and others' results, and discuss the reasons for the differences, It may be that our description of the current situation is too cumbersome, which makes you feel like this. We deleted the discussion as appropriate (pls see L.243-302).

Reviewer 4 Report

Authors in their text entitled " The development and application of Smart Dairy Farm System" provide interesting information about the use of smart technology and application in acquiring useful information about farming management practices. There are some missing information especially when describing methodology, that should be covered before any further consideration. Kindly see my comments bellow:

l. 14 DHI specify the abbreviation

l. 15 omit second full-stop

l 111 Authors refer to cluster analysis.  Authors should specify how was conducted and also include the results into the respective section.

l. 117-126 how mastitis was verified? Did you examined the animals? How man animals finaly presented mastitis?

l. 121 close to normal distribution. what do you mean? provide results of normality and skwewnes test.  Did the data finally had a normal distribution?

l 144 specify also what P stands for

l. 170-182 move this section to MM

l 183-184 specify clearly the two applications.

l. 197 authors refer to effect of pasture group but there in no mention about this in MM

l. 203 please specify in MM how GHG were estimated. Each animal had a sensor? Sensors measured the gasses daily for each pen? Where did the sensors were put in the pen? Generally, authors could describe all the materials and sensors used in the study.

l 224-240 please reduce the length of this section as many of the information has already been reported.

l 264 I think that you refer to net average values of methane emissions and not to intensities. Therefore, please specify why emissions were  decreased.

l 270-273 please rephrase because this contradicts with your data. it seems that your sensor did not work.

l. 274  Using ROC analysis in not a novel approach for predicting the occurrance of a disease. In addition, I am not so sure that it can be deternined as smart because the whole process is conducted manually and not by artificial intelligence tools. How possible is the incorporation of such algorithm in a smart application. Authors should also discuss the limitation of the study. It gives the sense that they used data derived from smart applications just two test two hypothesis/ problems that of grouping not grouping and mastitis control.

Author Response

Response: First of all, the authors would like to express our thanks to the Academic Editor, and many thanks also to the reviewers for the valuable comments. Give us the chance to improve our manuscript. We really appreciate it.

Please see the following replies to the reviewer’s comments:

Comments and Suggestions for Authors

Authors in their text entitled " The development and application of Smart Dairy Farm System" provide interesting information about the use of smart technology and application in acquiring useful information about farming management practices. There are some missing information especially when describing methodology, that should be covered before any further consideration. Kindly see my comments bellow:

  1. 14 DHI specify the abbreviation

Response: The full spelling of DHI has been added specified (pls see L.15 DHI: Dairy Herd Improvement).

2.l. 15 omit second full-stop

Response: The second full-stop has been deleted (pls see L.18).

3.l 111 Authors refer to cluster analysis. Authors should specify how was conducted and also include the results into the respective section.

Response: Cluster analysis mainly uses the clust function in the stats package of R language for analysis. The factors (milk production, parity, DIM, MP, ME, etc.) that need to be considered are put into the function for calculation, and finally the grouping results are obtained (pls see L.190).

We only group cattle better through cluster analysis, and the main purpose is to compare milk yield and greenhouse gases of cows after grouping. Therefore, the paper focuses on the explanation of results rather than the demonstration of methods.

4.l. 117-126 how mastitis was verified? Did you examined the animals? How man animals finaly presented mastitis?

Response: The cows was classified according to the international convention using somatic cell count (SCC) of DHI data, when SCC ≤ 200,000 /ml, the cow is considered healthy, when 200,000 /ml < SCC ≤ 500,000 /ml, the cow is considered subclinical mastitis, when SCC > 500,000 /ml, the cow is considered having clinical mastitis. In this paper, both subclinical and clinical mastitis were named as the mastitis risk group. (It is supplemented in L.137-149 of the article, and add citation)

5.l. 121 close to normal distribution. what do you mean? provide results of normality and skwewnes test. Did the data finally had a normal distribution?

Response: Due to the skewed distribution of the SCC values and the heterogeneous variance, SCC were first transformed into somatic cell score (SCS), which was belong to normal distribution, before the subsequent analysis. (pls see L.144)

6.l 144 specify also what P stands for

Response: It has been added in the article: P represents the probability of positive results; 1-P represents the probability of non-positive results (L.170)

7.l. 170-182 move this section to MM

Response: The position has been moved as required (pls see L.87-106).

8.l 183-184 specify clearly the two applications.

Response: Two applications: Nutrient grouping and Mastitis prediction (pls see L.106)

9.l. 197 authors refer to effect of pasture group but there in no mention about this in MM.

Response: The control groups were assigned into 9 pens with 30 cows each pen according to original grouping (Divided according to milk production of cows) (pls see L.128).

10.l. 203 please specify in MM how GHG were estimated. Each animal had a sensor? Sensors measured the gasses daily for each pen? Where did the sensors were put in the pen? Generally, authors could describe all the materials and sensors used in the study.

Response: greenhouse gases dataset in this paper were estimated using NDS software explained in MM . GHG collected by sensors didnot used as various condition and need to more precise (pls see L.131-135).

11.l 224-240 please reduce the length of this section as many of the information has already been reported.

Response: Thank you for your comments. It has been rearranged and written according to the requirements (pls see L.236-241).

12.l 264 I think that you refer to net average values of methane emissions and not to intensities. Therefore, please specify why emissions were  decreased.

Response: Cows were divided into groups to feed different diets. The prediction results of CNCPS system showed that after grouping, the nutrition in the feed was more suitable for the needs of cows, and the cows made better use of dietary nitrogen and , so the methane emissions of cows after grouping were reduced (pls see L.248-251; and pls see L.256-258).

13.l 270-273 please rephrase because this contradicts with your data. it seems that your sensor did not work.

Response: Thank you for your comments. Since we have explained this problem in the article, the explanation in this part has been deleted (pls see L.131-135).

  1. 274 Using ROC analysis in not a novel approach for predicting the occurrance of a disease. In addition, I am not so sure that it can be deternined as smart because the whole process is conducted manually and not by artificial intelligence tools. How possible is the incorporation of such algorithm in a smart application. Authors should also discuss the limitation of the study. It gives the sense that they used data derived from smart applications just two test two hypothesis/ problems that of grouping not grouping and mastitis control.

Response: Thank you for your comments. Although ROC curve is not the latest prediction method, the purpose of this paper is to make full use of farm data, interpret the hidden value of data, and further promote the application of intelligent system in farm production management. The “smart” intended to facilate the farm management and make the dairy production more environment-friendly and management easily. The SDFS would collect as more data as possible. The technicians and SDFS would help farmers to better use the data.

Up to date, most research about mastitis prediction focused on farm-year-month (pls see L.274). Through logistic regression and ROC curve description, we could find the risk indicators of dairy mastitis for individal cows. However this indicators would be different to various farms and lactation months. Further research is needed to make wider use of SDFS with regard to mastitis prediction. 

The limitations of this study have been added to the article (L.296; L.317).

Round 2

Reviewer 4 Report

Authors improved the quality of their text adding the majority of my comments in their revised manuscript and I feel that meets the standards of the journal. Therefore,  I recommend the further publication of the article  without any other change.